# Horticultural Activities Participation and College Students' Positive Mental Characters: Mediating Role of Academic Self-Efficacy

**Siyuan Guo** [1,2], **Tongyu Li** [1,2,*], **Binxia Xue** [1,2,*] and **Xiuxian Yang** [3]

1 School of Architecture, Harbin Institute of Technology, Harbin 150006, China
2 Key Laboratory of Cold Region Urban and Rural Human Settlement Environment Science and Technology, Ministry of Industry and Information Technology, Harbin Institute of Technology, Harbin 150001, China
3 Psychological Science and Health Management Center, Harbin Medical University, Harbin 150081, China
* Correspondence: li_tonghe@126.com (T.L.); binxia68@126.com (B.X.); Tel.: +86-0451-86281083 (T.L.); +86-0451-86281137 (B.X.)

**Abstract:** In recent years, the ongoing impact of the COVID-19 epidemic, irregular closed school life and frequent online teaching have negatively impacted the mental health and academic performance of many college students. Doing horticultural activities is an effective way to promote physical and mental health and enhance academic performance. This paper explores the relationship between horticultural activities participation, academic self-efficacy and positive mental characters under the perspective of disciplinary integration, with a view to promoting the mental health status and academic performance of college students and the application of horticultural therapy on college campuses. Questionnaires such as the Positive Mental Characters Scale for Chinese College Students (PMCS-CCS) and Academic Self-Efficacy Scale (ASES) are used to investigate 160 college students from four universities in China. The results show that horticultural activity participation is significantly positively correlated with academic self-efficacy ($r = 0.345$; $p < 0.01$) and positive mental characters ($r = 0.298$; $p < 0.01$), and horticultural activity participation can positively affect positive mental characters ($B = 0.135$, $p < 0.01$). At the same time, academic self-efficacy has a partial mediating effect between horticultural activity participation and positive mental characters. Universities can actively carry out campus horticultural activities to enhance students' horticultural activity participation, which in turn promotes academic self-efficacy and further enhances the positive psychological level of college students.

**Keywords:** horticultural activity; positive psychology; academic self-efficacy; mediating role; college students

## 1. Introduction

The college student group is a relatively special group, in the stage of transition from adolescence to adulthood. They need to complete many tasks such as gaining independence from their families of origin, establishing self-development goals and dealing with interpersonal relationships, thus facing many pressures in study, employment and socialization. In recent years, the mental health problems of Chinese university students have been gradually highlighted by the recurring effects of the COVID-19 pandemic [1]. Ma Z. et al. surveyed 746,217 Chinese university students during the COVID-19 pandemic and found that approximately 45% of them had mental health problems, with prevalence rates of 34.9%, 21.1% and 11.0% for acute stress, depression and anxiety symptoms, respectively [2]. There is also an increasing trend of college students becoming suicidal or endangering others due to depression [3]. In addition, during the COVID-19 pandemic, long hours of online teaching led to weak teacher–student interactions, and some students experienced a variety of problems such as poor concentration, lower learning efficiency and reduced interest in learning [4–6]. These have adversely affected the development of college students;

therefore, the mental health condition and learning effectiveness performance of college students have become the key topics of concern for college education in the post-epidemic era.

Positive mental character is an important research component of positive psychology and an important indicator of mental health. The founder of positive psychology, Seligman, conducted the "Values in Action" (VIA) project to collect and explore the character strengths and virtues of individuals [7–9]. After years of research, they identified 24 widely recognized positive psychological qualities and classified them into six dimensions: wisdom and knowledge, courage, humanity, justice, temperance and transcendence [10]. Based on the existing positive psychology theory, Chinese scholar Meng Wanjin (2009) conducted a study on positive mental character based on the national conditions of China and the psychological development characteristics of college students [11,12] and identified twenty positive mental characters under six dimensions for Chinese college students (Table 1), which laid the foundation for the exploration of positive mental characters among Chinese college students. Seligman believes that virtues and strengths are at the core of an individual's positive qualities, acting as buffers that can be beneficial weapons in overcoming mental illness. Positive mental characters stimulate the potential of individuals, increase mental toughness and motivate people to reach their highest values in life. Positive mental characters have two roles: first, they can have a positive relationship with positive developmental outcomes and can positively predict subjective well-being and quality of life [13]; second, they can have a negative relationship with negative developmental outcomes and can treat psychological problems, alleviate depressive symptoms and reduce risky behaviors [14,15]. Therefore, the cultivation of positive mental characters can help to solve the mental health problems of college students, and it is important to further explore the influencing factors and mechanisms of the effects of positive mental characters.

**Table 1.** Dimension and content of positive mental characters of Chinese college students.

| Dimension | Content |
| --- | --- |
| 1. Wisdom and knowledge | Creativity, curiosity, love of learning, thinking and observation |
| 2. Courage | Sincerity, courage and persistence, enthusiasm |
| 3. Humanity | Feel love, love and kindness, social intelligence |
| 4. Justice | Team spirit, integrity and fairness, leadership |
| 5. Temperance | Tolerance, modesty, prudence, self-control |
| 6. Transcendence | Touch of heart, hope and faith, humor |

Horticultural activities are an effective way to maintain and restore people's physical and mental functions and improve their quality of life through plants, their growing environment and various activities related to plants [16]. During the COVID-19 pandemic, horticultural activities were considered to be an easily achievable public health strategy [17], promoting active contact and interaction with nature, enhancing physical dexterity and hand-eye coordination and having numerous positive effects on people's physical and mental health [18,19]. In the field of positive psychology research, performing horticultural activities has been shown to help improve cognitive function, relieve stress, reduce anxiety and enhance well-being [20,21]. Collective horticultural activities can promote communication and exchange between people, and Waliczek et al. found that participation in horticultural activities can improve interpersonal relationships [22], which can help promote the humanity dimension of positive mental characters. In addition, satisfaction and pride are enhanced after successfully cultivating plants. Raymond et al. found that participation in horticultural activities increases self-esteem and creates the courage to do things differently in life [23], which contributes to the development of the courage and transcendence dimensions of positive mental characters. Zhong et al. found that the frequency of participation in indoor horticultural activities had a positive and small direct effect on psychological well-being ($\beta = 0.15$, BCa 95% CI of 0.03 to 0.25, $f^2 = 0.02$), and that the frequency of participation in outdoor horticultural activities had a positive and small direct effect on psychological well-being ($\beta = 0.16$, BCa 95% CI of 0.06 to 0.25, $f^2 = 0.02$) [24].

All these reveal that participation in horticultural activities may have a positive effect on positive mental characters. Therefore, this study proposes the hypothesis that:

**H1.** *Horticultural activities participation positively affects positive mental characters.*

Academic self-efficacy is an individual's judgment of whether he or she can successfully achieve a particular academic goal, reflects the individual's self-confidence in his or her learning ability and represents the individual's level of effort in completing academic tasks [25]. Academic self-efficacy has a significant positive predictive effect on individuals' learning motivation, level of commitment to learning and academic performance [26]. Students with high academic self-efficacy are confident that they can successfully complete their academic tasks, and when faced with academic difficulties and setbacks, they are able to work hard and stay engaged in their studies according to the academic goals they have set and reduce the impact of negative emotions directed at themselves due to academic setbacks. Academic self-efficacy and positive mental characters have been shown to be closely related. Wu J et al. found that academic self-efficacy can positively affect the self-esteem and subjective well-being of college students [27]. Medrano et al. found a positive correlation between academic self-efficacy and positive emotions in college students [28]. Kim et al. found that academic self-efficacy as the independent variable had a statistically significant effect on self-transcendence ($B = 0.15$, $p < 0.001$) [29]. Gulsen et al. found a significant positive correlation between academic self-efficacy and self-leadership ($r = 0.40$; $p < 0.01$) [30]. Considering that self-transcendence and self-leadership are components of positive mental characters, we believe that academic self-efficacy may positively affect positive mental characters.

Many researchers have demonstrated that participation in school garden projects improves student learning [31]. Additionally, related studies have shown that participation in horticultural activities and academic self-efficacy are closely related. Academic self-efficacy is described as an individual's confidence in his or her ability to hold a role and succeed in the educational context [32]. Ruiz-Gallardo et al. found that after a two-year garden-based learning program, students' skills and confidence in learning improved significantly [33]. For students of horticulture-related majors, participation in horticultural activities is a good practical opportunity to motivate students and exercise their problem-solving skills and confidence, thus helping them to better master their knowledge while promoting academic self-efficacy [34]. The numerous studies mentioned above suggest the possible mediating role of academic self-efficacy between horticultural activities participation and positive mental characters. Based on this, this study proposes the hypothesis that:

**H2.** *Academic self-efficacy has a mediating effect between horticultural activity participation and positive mental characters.*

In summary, this study explored the effects of horticultural activity participation on the positive mental characters of college students and the mediating role of academic self-efficacy between the two from a cross-disciplinary perspective, with the promotion of college students' mental health and academic performance as the starting point. It provides references and suggestions for the promotion of college students' mental health and academic competence, as well as the better application of horticultural therapy on college campuses.

## 2. Materials and Methods

### 2.1. Participants and Procedures

Convenience sampling is a non-probability sampling method in which respondents are randomly selected at a specific time and a certain position in a specific community in order to match the research theme. A convenience sampling method was used to conduct a questionnaire survey of 160 college students conveniently selected from four universities,

including Harbin Institute of Technology and Nanjing Forestry University, on a class basis. These universities have disciplines related to landscape architecture and have good campus horticultural activities (Figures 1 and 2). The questionnaires with response time less than 3 min and those with regular responses (i.e., the same option being selected repeatedly or answers in the pattern of 1, 2, 3, 4, 5) were excluded, and 147 valid questionnaires were recovered, with a valid recovery rate of 91.88%. The mean age of 147 subjects was 22.20 years old (SD = 2.13), including 66 males and 81 females, 91 undergraduates and 56 master's and doctoral students.

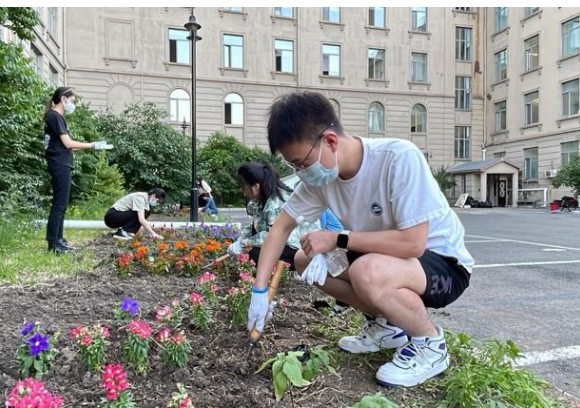

**Figure 1.** Horticultural activities on university campuses (1).

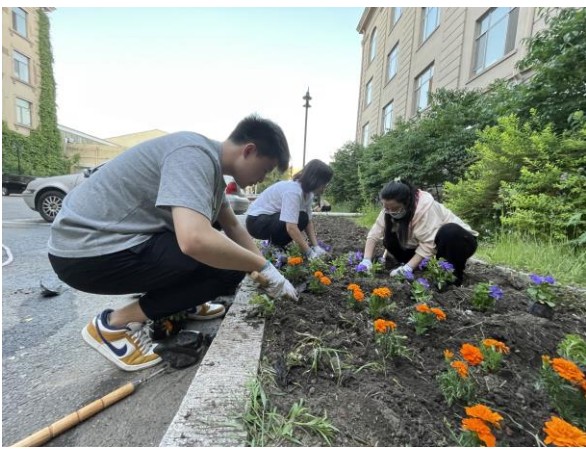

**Figure 2.** Horticultural activities on university campuses (2).

All participants participated voluntarily and could withdraw from the study at any time, and they were informed that all data were confidential and used only for research purposes. The ethical review and approval of the study was conducted by the Professor Committee of School of Architecture, Harbin Institute of Technology. The questionnaire took approximately 10 min to complete, and the data collection period was from 22–25 October 2022.

*2.2. Measures*

2.2.1. Horticultural Activity Participation

There is no specific scale to evaluate participation in horticultural activities, so this study draws on the indicators used to evaluate participation in sports activities [35,36] and volunteer activities [37], and sets five dimensions to evaluate participation in horticultural activities: frequency of participation, length of participation, continuous period, degree of investment and degree of achievement. Referring to Zhong et al., who used a Likert scale to evaluate the frequency of participation in horticultural activities among the research

participants in the past year [24], this study used a 5-point Likert scale to evaluate five dimensions of college students' participation in horticultural activities in the past year (Table 2). The Cronbach 's α for the overall scale was 0.920.

**Table 2.** Horticultural activity participation measurement dimensions and scoring rules.

| Dimension | Content | Score |
| --- | --- | --- |
| 1. Frequency of participation | Frequency of conducting horticultural activities | 1 = very low frequency to 5 = Very high frequency |
| 2. Length of participation | Length of time for each horticultural activity | 1 = Very short duration to 5 = Very long duration |
| 3. Continuous period | Period of continuous horticultural activity | 1 = Very short duration to 5 = Very long duration |
| 4. Degree of investment | Degree of investment when conducting horticultural activities | 1 = Very little investment to 5 = Very invested |
| 5. Degree of achievement | Degree of achievement of the expected results (e.g., successfully cultivate plants) | 1 = Never reached to 5 = Always reached |

### 2.2.2. Academic Self-Efficacy

We adopted the Academic Self-Efficacy Scale (ASES) developed by Liang (2000) to measure academic self-efficacy [38]. This scale refers to the academic self-efficacy questionnaire developed by Pintrich and DeGroot (1990) [39] and has two dimensions. The first 11 items relate to academic competence efficacy (i.e., students' confidence in successfully mastering academic subjects and achieving high scores), and the remaining 11 items relate to academic behavior efficacy (i.e., students' confidence in regulating their own studying and learning activities). Participants answer these 22 items on a 5-point Likert scale (ranging from 1 = completely disagree to 5 = completely agree). The total score of academic self-efficacy was the sum of the scores of the two dimensions, with higher scores representing a higher sense of efficacy. The ASES has been widely used in previous studies and has been shown to have good reliability and validity [40]. In our study, the Cronbach's α for the academic competence efficacy subscale and the academic behavior efficacy subscale were 0.752 and 0.742, respectively. The Cronbach's α of the whole scale was 0.847.

### 2.2.3. Positive Mental Characters

We used the Positive Mental Characters Scale for Chinese College Students (PMCS-CCS) developed by Meng and Guan (2009) [11], which is based on the theoretical framework and measurement tool of the Values in Action (VIA). It is a questionnaire with 62 items, which collects 20 positive mental characters distributed across the following six dimensions: wisdom and knowledge, courage, humanity, justice, temperance and transcendence. The response for each item was graded on a 5-point Likert scale: 1 = completely disagree to 5 = completely agree, with a higher score indicating a greater presence of the corresponding positive mental character. The PMCS-CCS has been widely used in previous studies and has been shown to have good reliability and validity [41]. In our study, the Cronbach's α for the overall scale was 0.955 and the Cronbach's α for each dimension was 0.710–0.847, which also had good reliability.

### 2.3. Data Analysis

We used SPSS software (version 22.0, IBM, New York, NY, USA) to process the data. First, the independent sample T test was used to compare the differences in positive mental characters of college students with different genders and educational backgrounds, and the correlations among the variables were explored through correlation analysis. Second, linear regression analysis was used to explore the relationships among horticultural activity participation, academic self-efficacy and positive mental characters; ridge regression was used to explore the effects of the five dimensions of horticultural activity participation on

positive mental characters. Third, based on the relationship between the variables, we used the structural equation model (SEM) to construct a model of the relationship between the three variables of horticultural activity participation, academic self-efficacy and positive mental characters. Finally, we tested the mediating effect of academic self-efficacy via bootstrap analysis.

According to the calculation of G*Power 3.1, the minimum number of subjects required for correlation analysis was 82 with tail = two, power $(1 - \beta) = 0.80$, $\alpha = 0.05$ and effect size = 0.30, and the effective sample size of this study met this requirement. After that, an independent sample T test and regression analysis were conducted a priori analysis, and the sample size was calculated. The results showed that the minimum subject size was lower than the effective subject size of this study, and the sample size of this study met the requirements.

## 3. Results

### 3.1. Findings Related to Independent Sample T Test

Independent sample T-test was used to analyze the positive mental characters of college students of different genders and educational backgrounds, and the results showed (Table 3) that there was no significant difference in the positive mental characters of students of different genders in the mean values of six dimensions and total scores. The positive mental characters of students with different educational backgrounds showed significant differences in wisdom and knowledge, courage, transcendence and total score average $(p < 0.05/p < 0.01)$. The positive mental characters of students with master's and doctoral degrees were significantly lower than those of undergraduates.

**Table 3.** Positive mental characters of different groups of college students.

| | Wisdom and Knowledge | Courage | Humanity | Justice | Temperance | Transcendence | Total Score Average |
|---|---|---|---|---|---|---|---|
| Gender—Male ($n$ = 66) | 3.76 ± 0.58 | 3.84 ± 0.56 | 3.78 ± 0.67 | 3.71 ± 0.72 | 3.64 ± 0.55 | 3.83 ± 0.57 | 3.76 ± 0.53 |
| Gender—Female ($n$ = 81) | 3.69 ± 0.54 | 3.76 ± 0.49 | 3.83 ± 0.56 | 3.73 ± 0.55 | 3.61 ± 0.52 | 3.77 ± 0.50 | 3.73 ± 0.46 |
| $t$ | −0.684 | −1.004 | 0.527 | 0.164 | −0.277 | −0.774 | −0.357 |
| $p$ | 0.495 | 0.317 | 0.599 | 0.870 | 0.782 | 0.440 | 0.721 |
| Educational Background—Undergraduate ($n$ = 91) | 3.84 ± 0.57 | 3.87 ± 0.50 | 3.88 ± 0.61 | 3.78 ± 0.58 | 3.67 ± 0.56 | 3.88 ± 0.54 | 3.82 ± 0.49 |
| Educational Background—Master and Doctor ($n$ = 56) | 3.53 ± 0.48 | 3.67 ± 0.53 | 3.69 ± 0.60 | 3.63 ± 0.70 | 3.55 ± 0.46 | 3.66 ± 0.50 | 3.62 ± 0.47 |
| $t$ | 3.366 | 2.378 | 1.801 | 1.344 | 1.281 | 2.492 | 2.453 |
| $p$ | 0.001 ** | 0.019 * | 0.074 | 0.181 | 0.202 | 0.014 * | 0.015 * |

Note: * $p < 0.05$, ** $p < 0.01$.

### 3.2. Findings Related to Correlation Analysis

The variables were analyzed using Pearson correlation analysis, and the results are shown in Table 4. Horticultural activity participation was significantly positively correlated with academic self-efficacy ($r$ = 0.345; $p < 0.01$) and positive mental characters ($r$ = 0.298; $p < 0.01$). There was also a significant positive correlation between academic self-efficacy and positive mental characters ($r$ = 0.321; $p < 0.01$).

**Table 4.** Results of Pearson correlation analysis.

| | Horticultural Activity Participation | Academic Self-Efficacy | Positive Mental Characters |
|---|---|---|---|
| Horticultural activity participation | 1 | | |
| Academic self-efficacy | 0.345 ** | 1 | |
| Positive mental characters | 0.298 ** | 0.321 ** | 1 |

Note: ** $p < 0.01$.

### 3.3. Findings Related to Regression Analysis

Regression analysis was conducted on the variables and the results are shown in Table 5. Horticultural activity participation can positively affect academic self-efficacy

($B$ = 0.142, $p$ < 0.01) and positive mental characters ($B$ = 0.135, $p$ < 0.01), while academic self-efficacy can positively affect positive mental characters ($B$ = 0.351, $p$ < 0.01). When academic self-efficacy was included in the model as a mediating variable, horticultural activity participation could still positively affect positive mental characters ($B$ = 0.096, $p$ < 0.05).

**Table 5.** Results of regression analysis and structural routes between variables.

|  | Independent Variable | Dependent Variable | $B$ | $t$ |
|---|---|---|---|---|
| Direct effect | Horticultural activity participation | Academic self-efficacy | 0.142 | 4.422 ** |
|  | Horticultural activity participation | Positive mental characters | 0.135 | 3.754 ** |
|  | Academic self-efficacy | Positive mental characters | 0.351 | 4.077 ** |
| Indirect effect | Horticultural activity participation | Positive mental characters | 0.096 | 2.583 * |

Note: * $p$ < 0.05, ** $p$ < 0.01.

To avoid the problem of collinearity, the scores on the five dimensions of horticultural activity participation were used as independent variables and the scores on positive mental characters were used as dependent variables for ridge regression analysis, with the K value taken as 0.990. The results in Table 6 show that the model passed the *F*-test ($F$ = 2.689, $p$ = 0.024 < 0.05) and that the model is significant. The length of participation in horticultural activity can significantly positively affect positive mental characters ($B$ = 0.028, $p$ < 0.05).

**Table 6.** Results of ridge regression analysis.

|  | $B$ | SE | *Beta* | $t$ | $p$ |
|---|---|---|---|---|---|
| 1. Frequency of participation | 0.017 | 0.013 | 0.045 | 1.305 | 0.194 |
| 2. Length of participation | 0.028 | 0.012 | 0.075 | 2.369 | 0.019 * |
| 3. Continuous period | 0.028 | 0.016 | 0.066 | 1.773 | 0.078 |
| 4. Degree of investment | 0.016 | 0.012 | 0.042 | 1.381 | 0.170 |
| 5. Degree of achievement | 0.018 | 0.013 | 0.044 | 1.401 | 0.163 |
| $R^2$ |  |  | 0.087 |  |  |
| Adjusted $R^2$ |  |  | 0.055 |  |  |
| $F$ |  | $F$ (5, 141) = 2.689, $p$ = 0.024 |  |  |  |

Note: * $p$ < 0.05.

### 3.4. Findings Related to Structural Equation Model

The structural equation model between the three variables was constructed based on the influence relationship between the variables (Figure 3). The model is well fitting ($\chi^2$/df = 1.885; RMSEA = 0.078; RMR = 0.028; GFI = 0.889; CFI = 0.962; NFI = 0.923; IFI = 0.963). The structure of Figure 3 shows that both horticultural activity participation ($\beta$ = 0.183, $p$ < 0.05) and academic self-efficacy ($\beta$ = 0.315, $p$ < 0.01) had a significant positive effect on positive mental characters and that horticultural activity participation had a significant positive effect on academic self-efficacy ($\beta$ = 0.370, $p$ < 0.01). This indicates that academic self-efficacy partially mediates the relationship between horticultural activity participation and positive mental characters.

### 3.5. Results of Mediating Effect Test

The mediating effect of academic self-efficacy was tested via bootstrap analysis. The results in Table 7 show that the mediating effect value of academic self-efficacy is 0.039, and the 95% confidence interval is 0.025–0.159, which does not include 0, indicating that the mediating effect is significant. The mediating effect accounted for 28.89% of the total effect, indicating that the mediating effect explained 28.89% of the relationship between participation in horticultural activities and positive mental characters.

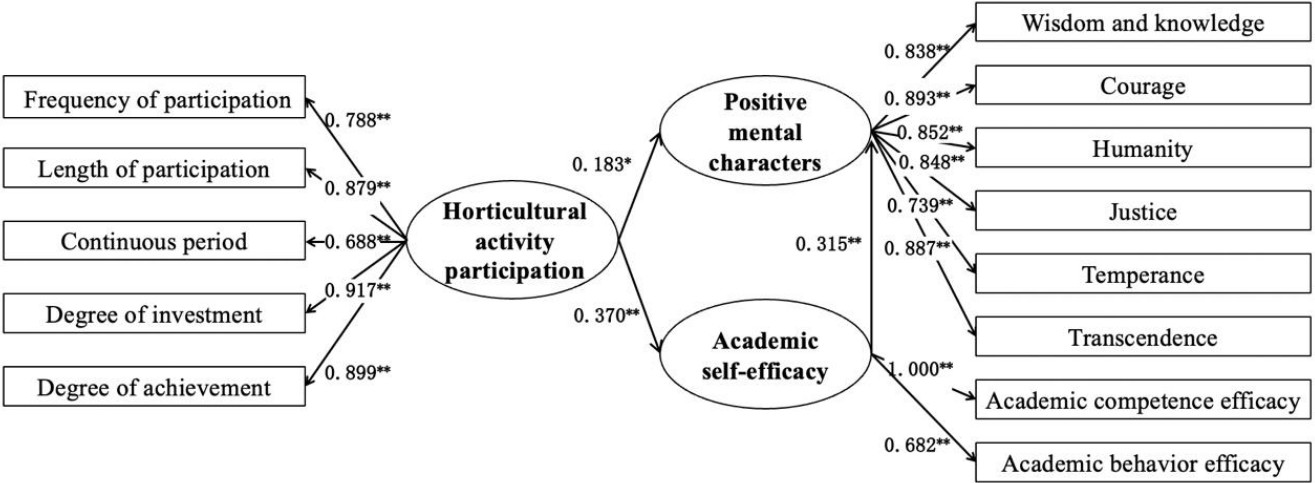

**Figure 3.** Model diagram of the relationship between variables. Note: * $p < 0.05$, ** $p < 0.01$.

**Table 7.** Effect Size and Bootstrap 95% CI.

| | Effect Size | Bootstrap 95% CI | | Relative Effect Size |
|---|---|---|---|---|
| | | Lower Limit | Upper Limit | |
| Total effect | 0.135 | 0.064 | 0.205 | |
| Direct effect | 0.096 | 0.023 | 0.169 | 71.11% |
| Mediating effect | 0.039 | 0.025 | 0.159 | 28.89% |

Note: Bootstrap 95% Confidence Interval (Bootstrap 95% CI); Relative Effect Size = Direct effect size or mediating effect size divided by total effect size.

## 4. Discussion

### 4.1. Relationship between Horticultural Activity Participation and Positive Mental Characters

According to the analysis results in Tables 4 and 5, this study found that there is a significant positive correlation between horticultural activity participation and positive mental characters and that horticultural activity participation can significantly positively affect positive mental characters, which verifies hypothesis 1. The following is the analysis of the effect of horticultural activity participation on different dimensions of positive mental characters.

For the wisdom and knowledge dimension of positive mental characters; horticultural activities are mainly divided into two types: productive (sowing, transplanting, etc.) and creative (flower arranging, bonsai arrangement, etc.) [42,43]. Creative horticultural activities contain many creative elements, and students can independently match plants and create landscapes, stimulating their creativity, curiosity and thinking [44] as well as promoting the wisdom and knowledge dimension of positive mental characters.

For the justice dimension of positive mental characters; gardening activities tend to be cooperative. In the various stages from sowing to harvesting, college students work to accomplish a common goal, help each other, and exchange emotions, promoting teamwork and leadership skills in the justice dimension of positive mental characters. This is consistent with the findings of Holmes et al., who found that students' teamwork and leadership skills were enhanced after gardening activities in campus gardens [45].

For the courage and humanity dimensions of positive mental characters; Pollin et al. defined the school garden as a social and emotional place [46], and in group horticultural activities, students socialize about plants, share planting experiences, experience the joy of harvesting and even share family and study problems with each other [47], all of which promote the courage and humanity dimensions of positive mental characters.

For the temperance dimension of positive mental characters; productive gardening activities emphasize participation and longevity, and the cultivation of living flowers

and trees requires caution and persistence in horticultural activities. For example, pruning flowers and trees should be cut selectively, and watering should be increased or decreased according to the different growth stages of plants. Therefore, long-term gardening helps to regulate impatient personalities and to develop prudence and self-control in the temperance dimension.

For the transcendence dimension of positive mental characters; after planting seeds, college students tend to be hopeful that the plants will take root and blossom and experience the joy of harvesting when the plants mature. This process can lead them to gain self-confidence, a sense of value and achievement and generate positive attitudes, which contribute to the transcendence dimension of positive mental characters.

In addition, according to the results in Table 6, among the five measured dimensions of horticultural activity participation, the length of participation had a significant positive effect on positive mental characters. This is consistent with the findings of Machida, who found that the time spent performing horticultural activities was significantly associated with people's health status, while the frequency of participation in horticultural activities was not significantly related to health status [48]. Gardening is a complementary therapeutic approach that requires a certain amount of time to accrue positive benefits. Angelia et al. found that those who gardened for less than one hour per week had much poorer mental resilience compared to those who gardened for longer periods of time per week [49]. Therefore, it is important to ensure that students have sufficient time to participate when planning campus horticultural activities.

### 4.2. Mediating Effect of Academic Self-Efficacy

After confirming the significant positive predictive effect of horticultural activity participation on positive mental characters, the study further explored the mediating role of academic self-efficacy in the two. Analysis of the mediating effect indicated that academic self-efficacy partially mediated the relationship between horticultural activity participation and positive mental characters. Horticultural activity participation not only directly predicted positive mental characters, but also influenced positive mental characters through the mediating effect of academic self-efficacy, which verified hypothesis 2.

Horticultural activity participation can significantly and positively affect academic self-efficacy. Long-term exposure to a high-pressure, fast-paced environment tends to make college students feel bad, while the process of horticultural activities is a slow-paced state, and engaging in horticultural activities helps reduce mental stress, which can better relieve students' academic stress and improve focus on the combination of work and rest as well as study efficiency and academic performance [50–52]. For the academic competence efficacy dimension of academic self-efficacy, persistent horticultural activities are conducive to the cultivation of persistence and patience, which build confidence for college students to better engage in learning, thus promoting academic competence efficacy. For the academic behavior efficacy dimension of academic self-efficacy, horticultural activities require comprehensive consideration of various factors such as soil, seeds, planting tools, irrigation water sources and sowing seasons, which helps college students develop the habit of making reasonable plans. For students in landscape architecture, horticulture, ecology and other related disciplines, performing horticultural activities also allows them to take the initiative to relate their knowledge from books and apply it to practice [53]. These have a positive impact on academic behavior efficacy.

While horticultural activity participation promotes academic self-efficacy, academic self-efficacy positively influences positive mental characters. On the one hand, students with high academic self-efficacy tend to achieve excellent grades [54], and excellent grades can lead to positive emotions and subjective well-being [55], promoting positive mental characters. On the other hand, the effects of high academic self-efficacy can be extended to other aspects in life. Students with high academic self-efficacy tend to have better psychological qualities, they believe that they can solve some problems independently and can face difficulties correctly [56]. Therefore, they are prone to develop self-confidence,

optimism and hope in life [57]. In addition, Zysberg et al. found that academic self-efficacy can facilitate interpersonal relationships [58]. These benefits enhance the formation of positive mental characters.

### 4.3. Suggestions for Promoting Positive Mental Characters through Horticultural Activities

College campus horticultural activities are a new way of education in China. According to the conclusion of the article, from the three levels of school, associations and students, we put forward suggestions to promote positive mental characters through horticultural activities, aiming to provide practical guidance for promoting college students' mental health and better promote the application and popularization of horticultural therapy in college education.

#### 4.3.1. School: Stable Horticultural Support

To increase positive experiences and form a positive personality requires an external environment system as a supporting condition. In the study of positive psychology, the external environment that enables individuals to obtain more positive experiences and is easy to form positive personality is called a positive environmental system. The university campus is an important part of the positive social organization system [59]. Universities and colleges can try to build a positive and stable support system for horticultural activities, so as to create basic conditions for college students to carry out horticultural activities and improve their positive mental characters.

According to the analysis results in Figure 3 and Table 7, academic self-efficacy plays a mediating role between participation in horticulture activities and positive mental characters, and linking horticulture therapy with college students' learning is an important way to promote positive mental characters. Schools can carry out some optional courses related to horticultural activities (such as flower arrangement and artistic creation, etc.). Based on relevant professional characteristics and educational concepts, colleges and universities can reasonably integrate horticultural activities into the teaching system, build a comprehensive horticultural education system including horticultural courses, horticultural competitions, horticultural practices and horticultural exhibitions and promote the sustainable development of horticultural therapy in colleges and universities. These measures increase students' understanding of knowledge and promote students' academic self-efficacy. In addition, schools can arrange gardening sites and coordinate activity time to form multi-directional horticultural support, so as to effectively guide college students to actively participate in campus horticultural activities and promote their mental health.

#### 4.3.2. Association: Sustainable Horticultural Activities

According to the analysis results in Table 6, the length of participation had a significant positive effect on positive mental characters ($B = 0.028$, $p < 0.05$). However, horticultural activities in the four universities investigated in this study are mostly short-term activities, which affects the effect of horticultural activities and damages students' interest and enthusiasm to some extent. According to the interview, the reason for the short duration of the activity is mainly the lack of collective long-term activity organization and the lack of corresponding management and maintenance in the later period of the activity.

Associations in universities not only gather like-minded friends, but also provide support for activities. The establishment of associations related to horticultural therapy is an important form of horticultural therapy application on campus and also an important measure to realize the specialization, systematization and continuation of the horticultural activity organization of college students. After the establishment of associations related to horticulture therapy, it is possible to cooperate with other administrative student associations such as the Volunteer Association and take advantage of their advantages in organizing activities and mobilizing the masses to jointly promote the sustainable development of campus horticultural activities in an organized way.

### 4.3.3. Student: Differentiated Horticultural Guidance

According to the results in Table 3, there are differences in positive mental characters between undergraduates, master's and PhD students, which is mainly caused by the different training modes and learning requirements of the two groups. Graduate students and doctoral students require higher abilities of innovation and research than undergraduates, and affected by age, they tend to experience greater pressures of employment, so they often have too much pressure and a lack of positive psychology [60,61]. Therefore, we need differentiated horticultural guidance for both groups. The pressure of undergraduates mainly comes from course learning, interpersonal communication, employment, etc. They often spend more time outside class, and horticultural activities can be an important way to enrich extracurricular activities. Through horticultural activities, they cultivate their abilities and promote social interaction so as to promote positive psychology. At the same time, students can give full play to their subjective initiative, become creative with the forms of the campus gardening activities, focus on the easy access of indoor space such as classrooms and dormitories, use plants for indoor greening and decoration, participate in horticultural practice through various ways and realize that the experiences of horticultural activities are not restricted by time, place and season.

However, master's and doctor's students usually have less extracurricular time, and the pressure mainly comes from scientific research and employment. Therefore, gardening can be taken as an effective way to relax. At the same time, for some masters and doctors of psychology, landscape architecture, public health and other majors, they can combine their own scientific research topics with horticultural activities, view participation in horticultural activities as an important way to collect data, observe user behavior, obtain scientific research resources, enrich scientific research results and enhance academic confidence. This can promote the academic self-efficacy of master's and doctoral students and further promote the positive mental characters of master's and doctoral students through the mediating effect of academic self-efficacy.

### 4.4. Limitations

There are some limitations in this study. On the one hand, the valid research subjects of this study are one hundred forty-seven college students from four universities, which cannot represent the overall college students in China. The study area and sample size can be expanded later for an in-depth study. On the other hand, there is a gap between the number of undergraduate students and the number of master's and PhD students in the research population of this study. There are differences between undergraduate students and master's and doctoral students in terms of psychological stress and academic requirements [62,63], so there may also be differences in the role of horticultural activity participation on academic self-efficacy and positive mental characters. A more in-depth study from a demographic perspective can be conducted later.

## 5. Conclusions

Based on theoretical knowledge from the disciplines of healing landscape, horticultural therapy and positive psychology, this study focused on a group of college students to explore the relationship between horticultural activities participation, academic self-efficacy and positive mental characters. The main conclusions are as follows:

(1) Horticultural activity participation was significantly positively correlated with academic self-efficacy ($r = 0.345$; $p < 0.01$) and positive mental characters ($r = 0.298$; $p < 0.01$). Academic self-efficacy was significantly positively correlated with positive mental characters ($r = 0.321$; $p < 0.01$).

(2) Horticultural activity participation can positively affect positive mental characters ($B = 0.135$, $p < 0.01$), and the length of participation in horticultural activity can significantly positively affect positive mental characters ($B = 0.028$, $p < 0.05$).

(3) Academic self-efficacy partially mediates the relationship between horticultural activity participation and positive mental characters, and horticultural activity participa-

tion not only directly predicts positive mental characters, but also has an impact on positive mental characters through the mediating role of academic self-efficacy.

In summary, horticultural activities, as physical and mental health interventions that emphasize natural elements and healing effects, have important significance in promoting college students' academic performance and mental health. Universities can reasonably integrate horticultural activities into the teaching system, create a comprehensive horticultural education system including horticultural courses, horticultural competitions and horticultural practices. In addition, universities can create student associations related to horticultural therapy. Through the above measures, we promote the diversification, specialization, systematization and sustainability of horticultural activities in university campuses. At the same time, appropriate horticultural guidance is provided to different groups in order to achieve a win-win situation of improving the physical and mental health of college students and promoting their learning performance.

**Author Contributions:** Conceptualization, S.G., T.L. and B.X.; methodology, S.G., T.L. and B.X.; software, S.G.; validation, X.Y.; formal analysis, S.G.; investigation, S.G.; resources, X.Y.; data curation, X.Y.; writing—original draft preparation, S.G.; writing—review and editing, T.L., B.X. and X.Y.; visualization, S.G.; supervision, T.L., B.X. and X.Y.; project administration, T.L. and B.X.; funding acquisition, T.L. and B.X. All authors have read and agreed to the published version of the manuscript.

**Funding:** This research was funded by Heilongjiang Provincial Natural Science Foundation of China (No. LH2021E068 and No. LH2020E052); the Opening Fund of Key Laboratory of Interactive Media Design and Equipment Service Innovation, Ministry of Culture and Tourism (No. 20201 and No. 20206); Harbin Institute of Technology Graduate Education Teaching Reform Research Project (No. 22MS036).

**Data Availability Statement:** Not applicable.

**Acknowledgments:** The authors are grateful to Li Jiayang of the college of Intelligent Science and Engineering, Shenyang University for her help with data analysis; and Anna María Pálsdóttir of the Department of People and Society, Faculty of Landscape Architecture, Horticulture and Crop Production Science, Swedish University of Agricultural Sciences for her help with the revision of the paper and editing in English. In addition, the authors are grateful to the editors and anonymous reviewers for their insightful comments and suggestions.

**Conflicts of Interest:** The authors declare no conflict of interest.

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
