# Peer review of "Horticultural Activities Participation and College Students’ Positive Mental Characters: Mediating Role of Academic Self-Efficacy"

_horticulturae, doi:10.3390/horticulturae9030334_

Round 1
Reviewer 1 Report
I found the subject matter of this article fascinating! The researchers did a nice job laying out the premise of the study and guiding the reader through the intricacies of the large amount of data. Please see below the few minor edits I identified:
Table 2. Very little investment to very invested
line 195. characters. Third
line 196. (SEM)
line 277. delete and impatient
Author Response
Dear Editors and Reviewers:
Thank you for your letter and for the reviewers' comments concerning our manuscript entitled " Horticultural Activities Participation and College Students' Positive Mental Characters: Mediating Role of Academic Self-efficacy" (ID: horticulturae-2236895). Those comments are all valuable and very helpful for revising and improving our paper, as well as the important guiding significance to our research. We have studied comments carefully and have made correction which we hope meet with approval. Revisions to the manuscript are marked up using the “Track Changes” function in the paper. The main corrections in the paper and the responds to the reviewer's comments are as flowing:
Responds to Reviewer 1 Comments:
1.Response to comment 1:Table 2. Very little investment to very invested
Response:
Thanks for the reviewer's comment. We have modified it in Table 2, and the modified content is "Very little investment to very invested".
2.Response to comment 2: line 195. characters. Third
Response:
We have modified it in line 240, and the modified content is " characters. Third ".
3.Response to comment 3: line 196. (SEM)
Response:
We have modified it in line 241, and the modified content is " (SEM) ".
4.Response to comment 4: line 277. delete and impatient
Response:
Thanks for the reviewer's suggestion. We have deleted " and impatient " in line 357.

Reviewer 2 Report
dear authors,
the paper entitled "Horticultural Activities Participation and College Students' Positive
Mental Characters: Mediating Role of Academic Self-efficacy" investigates a very important topic, namely a cost-effective and efficient form of intervention that may significantly improve mental health in a relatively understudied population: students.
I would like to drive attention to several aspects:
1. your investigation is a single assessment correlational study, with a relatively heterogenous group of participants (undergraduates, master and doctoral students), Consequently, one cannot even have a hint towards causal relationships between variables. One can only state that there are significant, strong ASSOCIATIONS, but cannot conclude that one variable affects another variable.
2. The heterogeneity of participants requires further attention in processing data and interpreting results, since there may be significant differences due to age and developmental levels between participants.
3. There also may be interesting to investigate the way in which variables relate to each other depending on categories: low, medium, and high levels of mental characteristics.
4. I would suggest presenting in more detail the Conclusion part and the practical implication of the results.
5. A thorough check of English is also highly recommended.
6. I would also suggest to describe the way the number of participants was calculated by using of g*power.
Author Response
Dear Editors and Reviewers:
Thank you for your letter and for the reviewers' comments concerning our manuscript entitled " Horticultural Activities Participation and College Students' Positive Mental Characters: Mediating Role of Academic Self-efficacy" (ID: horticulturae-2236895). Those comments are all valuable and very helpful for revising and improving our paper, as well as the important guiding significance to our research. We have studied comments carefully and have made correction which we hope meet with approval. Revisions to the manuscript are marked up using the “Track Changes” function in the paper. The main corrections in the paper and the responds to the reviewer's comments are as flowing:
Responds to Reviewer 2 Comments:
1.Response to comment 1:Your investigation is a single assessment correlational study, with a relatively heterogenous group of participants (undergraduates, master and doctoral students), Consequently, one cannot even have a hint towards causal relationships between variables. One can only state that there are significant, strong ASSOCIATIONS, but cannot conclude that one variable affects another variable.
Response:
Thanks for the reviewer's comment. We modified the introduction by adding references that reveal the influence relationship between variables (Line 96-100; Line 118-123). Based on some conclusions of other scholars, we predict that there may be an influence relationship between variables. In addition, the regression analysis is carried out in Section 3.3(Line 279-287), and the significant influence relationship is obtained.
2.Response to comment 2: The heterogeneity of participants requires further attention in processing data and interpreting results, since there may be significant differences due to age and developmental levels between participants.
Response:
Thanks for the reviewer's comment. We added section 3.1 to supplement these. In Section 3.1 (Line 256-266), independent sample T-test was used to analyze the positive mental characters of college students of different genders and educational backgrounds. In addition, according to the results of independent sample T-test, suggestions on horticultural activities for different groups of college students are added in section 4.3.3 of discussion (Line 459-485), so as to better promote mental health.
3.Response to comment 3: There also may be interesting to investigate the way in which variables relate to each other depending on categories: low, medium, and high levels of mental characteristics.
Response:
Thank you very much for the reviewer's advice. This is a very good perspective, and we tried to analyze it, but it may be that such a study requires a larger sample size, which we would like to further analyze in a follow-up study.
4.Response to comment 4: I would suggest presenting in more detail the Conclusion part and the practical implication of the results.
Response:
Thanks for the reviewer's suggestion. We added section 4.3 to supplement these. In Section 4.3 (Line 412-485), according to the conclusion of the article, from the three levels of school, associations and students, we put forward suggestions to promote positive mental characters through horticultural activities, aiming to provide practical guidance for promoting college students' mental health, and better promote the application and popularization of horticultural therapy in college education.
5.Response to comment 5: A thorough check of English is also highly recommended.
Response:
Thanks for the reviewer's suggestion. We have made some adjustments in English, and we plan to submit the manuscript to the English editing department of MDPI for English editing in the later stage.
6.Response to comment 6: I would also suggest to describe the way the number of participants was calculated by using of g*power.
Response:
Thanks for the reviewer's suggestion. We used G power to perform an a priori analysis of the sample size, and the sample size in the paper met the requirements, as described in line 245-254.

Reviewer 3 Report
1. In the research methods, more specific explanation on selection of participants is needed. What is a convenient sampling, and how the random survey was performed?
2. In the discussion, implication of the research should be suggested for students' mental or psychological health.
Author Response
Dear Editors and Reviewers:
Thank you for your letter and for the reviewers' comments concerning our manuscript entitled " Horticultural Activities Participation and College Students' Positive Mental Characters: Mediating Role of Academic Self-efficacy" (ID: horticulturae-2236895). Those comments are all valuable and very helpful for revising and improving our paper, as well as the important guiding significance to our research. We have studied comments carefully and have made correction which we hope meet with approval. Revisions to the manuscript are marked up using the “Track Changes” function in the paper. The main corrections in the paper and the responds to the reviewer's comments are as flowing:
Responds to Reviewer 3 Comments:
1.Response to comment 1:In the research methods, more specific explanation on selection of participants is needed. What is a convenient sampling, and how the random survey was performed?
Response:
Thanks for the reviewer's comment. Considering the reviewer's suggestion, we have added the definition of convenience sampling and the details of the survey process. (Line 156-165).
2.Response to comment 2: In the discussion, implication of the research should be suggested for students' mental or psychological health.
Response:
Thanks for the reviewer's suggestion. We added section 4.3 to supplement these. In Section 4.3 (Line 412-485), according to the conclusion of the article, from the three levels of school, associations and students, we put forward suggestions to promote positive mental characters through horticultural activities, aiming to provide practical guidance for promoting college students' mental health, and better promote the application and popularization of horticultural therapy in college education.

Round 2
Reviewer 2 Report
dear authors,
The new version of the manuscript has significantly improved. The only thing I would suggest is a final check for English.